# Trans-eQTLs Can Be Used to Identify Tissue-Specific Gene Regulatory Networks

**DOI:** 10.3390/cimb47080594

**Published:** 2025-07-29

**Authors:** Majid Nikpay

**Affiliations:** Omics and Biomedical Analysis Core Facility, University of Ottawa Heart Institute, Ottawa, ON K1Y 4W7, Canada; mnikpay@ottawaheart.ca

**Keywords:** trans-eQTLs, tissue specificity, gene regulatory network, mendelian randomization, scale-free network

## Abstract

Previous high-throughput screening studies have indicated that trans-eQTLs tend to be tissue-specific. This study investigates whether this feature can be used to identify tissue-specific gene regulatory networks. eQTL data for 19,960 genes were obtained from the eQTLGen study. Next, eQTLs displaying both cis- and trans-regulatory effects (*p* < 5 × 10^−8^) were selected, and the association between their corresponding genes was examined by Mendelian randomization. The findings were further validated using eQTL data from the INTERVAL study. The trans-regulatory impact of 138 genes on 342 genes was detected (*p* < 5 × 10^−8^). The majority of the identified gene-pairs were aggregated into networks with scale-free topology. An examination of the function of genes indicates they are involved in immune processes. The hub genes primarily shared transcription regulation activity and were associated with blood cell traits. The hub gene, *DDAH2*, impacted several metabolic and autoimmune disorders. On average, a gene in the network was under the regulatory control of 34 cis-eQTLs and 6 trans-eQTLs, and genes with higher heritabilities tended to exert higher regulatory impacts. This study reports tissue-specific gene regulatory networks can be detected by investigating their genomic underpinnings. The identified networks displayed scale-free topology, indicating that hub genes within a network could be targeted to correct abnormalities.

## 1. Introduction

The combination of active genes and their interactions within a cell generates a gene regulatory network (GRN) that enables the cell to specialize in fulfilling its function. Therefore, an improved understanding of GRNs not only paves the way to understand the principles of gene regulation but also serves as a tool to drive cell fate for purposes such as cellular engineering or to prevent disease outcomes [1].

Several approaches exist to infer GRNs from omics data [1,2,3]. These methods rely on a combination of biological and statistical information to identify related gene-pairs for GRN construction. Approaches such as co-expression network analysis aim to build a network by identifying genes whose expression profiles are correlated. While these methods systematically examine the genome and are thus hypothesis-free, the resulting network is undirected due to the symmetrical nature of correlations. Other methods attempt to resolve this issue by incorporating biological information. Notably, they utilize data from ChIP-seq and cis-regulatory elements to identify transcription factors that bind to target genes, thereby transforming an undirected network into one where edges indicate causality. However, the GRNs obtained from such approaches are limited by the availability of biological information. They assign transcription factors to their target genes based on genomic proximity, which could introduce bias in the presence of distal regulatory effects [1,2,3]; moreover, current methods are not protected against environmental factors that could cause confounding effects. Finally, existing approaches often rely on individual-level data, limiting their applicability in situations where data sharing is required. In this study, a method based on eQTLs is described that can address the aforementioned issues.

An eQTL or an expression quantitative trait locus is a site on DNA where variation in its sequence impacts the expression of a gene. If an eQTL is located near the gene it acts upon, it is referred to as a cis-eQTL; however, if it is located distant from its gene of origin, sometimes on a different chromosome, it is referred to as a trans-eQTL. Over the past decades, high-throughput studies have been conducted to map eQTLs. The results from these studies, which essentially summarize the magnitudes and natures of associations between genomic variants (eQTLs) and genes, are then collectively considered to investigate the genetics of the transcriptome. An important insight from these studies is the evidence that trans-eQTLs tend to be tissue-specific [4,5,6].

A recent development in this field is the advent of statistical methods that can leverage publicly available GWAS summary statistics including eQTLs to investigate the nature of the relationship between two biological entities (e.g., two genes) [7,8]. A prominent method in this regard is Mendelian randomization, which can not only test the association between two genes but also differentiate between causation and correlation [9,10]. Moreover, because Mendelian randomization uses a set of independent SNPs for association testing, its results are robust to confounding effects that could be introduced by environmental (non-genetic) factors. Building upon these advances, this study introduces a workflow based on Mendelian randomization that can complement previous approaches for constructing tissue-specific GRNs.

## 2. Methods

Figure 1 provides an overview of the workflow used in this study. The workflow requires two sets of independent eQTL data for discovery and validation purposes.

The analyses begin by identifying eQTLs that display both cis and trans effects (*p* < 5 × 10^−8^). This threshold was chosen because it has traditionally been used to declare an association significant in genome-wide studies. For each eQTL, Mendelian randomization is then performed to test whether a change in the expression of its source gene (under the cis effect) impacts the expression of its target gene (under the trans effect). Gene-pairs that show significant associations (*p* < 5 × 10^−8^) are selected, and their associations are further investigated at the validation step. Significant findings are then plotted to visualize their distributions and functional properties. The nature of the data used at the discovery and validation steps is described in the following sections.

### 2.1. Data for the Discovery Step

Previously, the eQTLGen study investigated the genetic architecture of blood transcriptome [11]. The data in this study consist of 31,684 blood and PBMC samples (19.6% of samples) obtained mainly from studies conducted in European populations. Gene expression profiling of samples was carried out using expression arrays and RNA sequencing (20.3% of samples). In the eQTLGen study, the authors combined eQTL data from 37 studies and reported the outcome of meta-analyses for 19,960 genes [11]. After processing these data, GWAS summary association statistics for 16,732 genes with at least one eQTL (*p* < 5 × 10^−8^) were obtained and used as the input data for the discovery analyses.

### 2.2. Data for the Validation Step

To replicate the findings from the discovery step, a second set of eQTLs was obtained from the INTERVAL study, in which the authors recently conducted eQTL mapping in blood samples from 4,732 participants of European origin [12]. The INTERVAL study is a randomized trial of healthy blood donors, who were recruited at England’s National Health Service Blood and Transplant center. Gene expression profiling of the samples was achieved by whole-blood Illumina RNA sequencing, and the authors provided GWAS summary association statistics for 17,362 genes. From this database, eQTLs for 15,298 genes, which are also reported in the eQTLGen study, were obtained.

### 2.3. Mendelian Randomization

To investigate the nature of relation between genes under the regulatory impact of an eQTL, Mendelian randomization was used.

Mendelian randomization is a statistical test that can investigate the nature of the relationship between two biological entities (e.g., two genes) by comparing their magnitudes of association to a group of shared SNPs (eQTLs in this study). The test uses a group of independent eQTLs (in linkage equilibrium, r^2^ < 0.2) that are associated (*p* < 5 × 10^−8^) with the expression of the source gene to investigate if change in the expression of the source gene impacts the expression of the target gene. In this study, I used the GSMR program to conduct the Mendelian randomization analysis [10]. The program tests the nature of association between the genes by plotting eQTLs on a scatter plot based on their effect sizes (β regression coefficients) on the source gene (x-axis) and the target gene (y-axis) and calculating the statistical parameters of the regression line. As compared to other methods for Mendelian randomization, GSMR accounts for the sampling variance in β estimates and the linkage disequilibrium (LD) among eQTLs, thus possessing higher statistical power [10].

Given that eQTL data were obtained from samples of European origin, genotype data from the 1000 Genomes-European population were used to compute the degree of linkage disequilibrium (LD) among SNPs and to match alleles. Following Mendelian randomization, gene-pairs that showed significant (*p* < 5 × 10^−8^) and concordant direction of association in both the discovery and validation steps were selected and plotted using the Cytoscape software (version 3.10.1) [13] to determine if they form a network. The DAVID functional tool (version 2023q4) [14] was used to identify biological processes that were overrepresented among the identified genes.

### 2.4. Phenom-Wide Association Study

To identify phenotypes associated with hub genes, a phenome-wide association study (PheWAS) was performed. This involved selecting independent eQTLs (r^2^ < 0.2) associated (*p* < 5 × 10^−8^) with a hub gene and examining their collective contribution to a phenotype through Mendelian randomization. To reduce the possibility of spurious associations, analyses were conducted using eQTL data from the eQTLGen study and then repeated using eQTL data from the INTERVAL study. Only gene-trait associations with *p* < 5 × 10^−8^ were considered significant. Analyses were performed using the GSMR program [10]. Genotype data from the 1000 Genomes-European population were used to compute the degree of LD among SNPs and to match alleles.

### 2.5. eQTL Assesment

To understand the characteristics of eQTLs underlying the identified genes, I generated a list of independent (r^2^ < 0.2) eQTLs per gene using the clump algorithm implemented in PLINK (v.1.9) [15]. In summary, the algorithm takes a list of eQTLs and their *p*-values, conducts LD pruning, and outputs a list of eQTLs in linkage equilibrium and prioritized by *p*-values. Following LD pruning, the phenotypic variance (
$V_{P}$, proportion of variation in a gene expression) attributed to an eQTL was calculated using the equation:
(1)$$V_{P}=2P\left(1-P\right)\beta^{2}$$ where
P is the frequency of minor allele and
β is its regression coefficient derived from the association model [16]. eQTLGen study reported Z-scores instead of regression coefficients. As such, a conversion was made using the equation:
(2)$$\beta =\frac{Z}{\sqrt{2F(1-F)(N+Z^{2})}}$$ where
Z represents Z-score,
F is the frequency of effect allele, and
N is the sample size [8].

## 3. Results

By following the analytical pipeline described in Figure 1, trans-eQTLs and cis-eQTLs were initially matched to identify eQTLs that display both cis- and trans-regulatory effects. Through this procedure, 55,884 gene-pairs were identified to share at least one eQTL (*p* < 5 × 10^−8^) in the eQTLGen dataset (discovery dataset). These findings were then investigated in the INTERVAL dataset (validation dataset), where a total of 15,522 gene-pairs were identified to also share an eQTL (*p* < 5 × 10^−8^). This was followed by performing Mendelian randomization and re-investigating the significant findings at the validation step. A total of 617 gene-pairs showed significant causal association (*p* < 5 × 10^−8^) in both the discovery and validation steps. The majority of these gene-pairs (*N* = 597) showed concordant direction of associations between the discovery and validation steps, and these were selected for post hoc analysis (Appendix A). The identified gene-pairs consisted of 138 genes that exerted trans-regulatory impact on 342 genes (a total of 474 unique genes, with 6 genes acting as both source and target gene).

The STRING database [17] provides functional interactions obtained from various biological resources. A search in this database was subsequently performed to investigate whether the findings from the Mendelian randomization (Appendix A) could be supported by existing biological knowledge. Supporting evidence for 32 gene-pairs was obtained from the STRING database (Appendix A). Furthermore, gene ontology (GO) enrichment analysis was conducted to determine if the identified genes (Appendix A) are functionally related. The findings indicated that the identified genes are involved in immune processes (Table 1).

To investigate possible bias in the findings that could be due to the nature of the eQTLGen study, 1000 gene-sets of the same size (*N* = 474 genes) were randomly drawn from the eQTLGen dataset, and their functions were investigated. This analysis was performed in the R environment (version 4.2.2) using gprofiler2, which systematically performs GO enrichment analysis on input gene lists [18]. The outcome of these analyses was mainly null, revealing only a few unrelated biological processes (Appendix A). Furthermore, to investigate the possibility of bias in the results due to the presence of the HLA region, genes located within this region (coordinates: chr6:28,477,797-33,448,354, based on GRCh37) were excluded, and the remaining gene-pairs were subjected to functional enrichment analysis. The findings once more confirmed that the identified gene-pairs share immune function (GO Term: 0045087, innate immune response, Bonferroni-corrected *p* = 0.003).

To investigate the distribution of the identified gene-pairs, they were entered into the Cytoscape software, which attempts to generate a network from a list of gene-pairs by connecting pairs that share a gene. The analysis revealed that the majority of gene-pairs aggregated into networks with scale-free topology (Figure 2). The distribution of the number of genes by frequency of edges also indicated a power-law distribution (Appendix A). While 77% of genes (*N* = 361) had on average 1.3 edges, a group of 10 genes (*ENSG00000267074*, *AP2B1*, *CREB5*, *DDAH2*, *LCN2*, *IKZF1*, *NFKBIA*, *NFE2*, and *PPP2R3C*) had an average of 24 edges and accounted for 20% of interactions. These hub genes had a high regulatory impact on the network. Notably, among them was *ENSG00000267074*, which impacted 85 genes. Examination of the function of these genes indicates their involvement in blood coagulation processes. *ENSG00000267074* is a long non-coding RNA (lncRNA) gene; therefore, its trans-regulatory function could be attributed to its role in transcription regulation. *CREB5*, *IKZF1*, *NFKBIA*, *NFE2*, and *PPP2R3C* were other hub genes that also had transcriptional regulation functions.

Next, a Phenome-Wide Association Study (PheWAS) was performed to examine the impact of the hub genes on the phenome. Findings from the discovery and validation analyses showed that hub genes are mainly involved in regulating blood cell traits (Appendix A). Furthermore, the hub gene *DDAH2* displayed pleiotropic impact on several metabolic and autoimmune conditions (Table 2). Higher expression of this gene had a favorable metabolic impact, reducing the risk of dyslipidemia and type 2 diabetes; however, its impact on autoimmune conditions varied. While it increased the risk of multiple sclerosis and rheumatoid arthritis, it was associated with lowered risk of type 1 diabetes, Crohn’s disease, primary biliary cholangitis, and psoriasis.

In contrast to the hub genes, I identified genes that were the target of several genes. Notably, *TRBV4-1*, *RNF5P1*, *TRAV26-2*, *TRBV7-3*, *TRAV35*, *GZMK*, *TRAV20*, *TRAV38-1*, *TRBV20-1*, and *CD248* were under the regulatory impact of ≥5 genes. Examination of the function of these genes indicated they share immune function and were mainly under the regulatory impact of genes in the HLA region. TRAV20, TRAV26-2, TRAV35, TRAV38-1, TRBV20-1, TRBV4-1, and TRBV7-3 are T cell receptors; moreover, CD248 and GZMK also have immune functions.

The characteristics of the eQTLs underpinning the identified genes were then examined. On average, a gene was under the regulatory impact of 34 cis-eQTLs (SE = 1.7, Median = 23) and 6 trans-eQTLs (SE = 0.3, Median = 4). This indicates that Mendelian randomization is well suited to identify functional interactions. On average, a cis-eQTL explained 1% of the variance in the expression of a gene, while this value was 0.6% for a trans-eQTL. Genes in the network that tended to act as source genes displayed higher gene expression heritability compared to those acting as target genes (*p* = 5.9 × 10^−6^). Furthermore, these genes had higher centrality measures, including betweenness centrality (*p* = 0.03) and closeness centrality (*p* = 2.2 × 10^−16^).

## 4. Discussion

A gene regulatory network (GRN) enables a cell to specialize in carrying out its function. The human body contains various cell types, and identifying their GRNs is crucial for various biological purposes, including improving disease diagnosis and treatment. Currently, several approaches can investigate such networks by analyzing raw individual-level data. This reliance on raw data hinders collaboration among researchers due to patients privacy and logistical considerations in data sharing. Furthermore, as discussed in the introduction, existing methods make several assumptions for generating GRNs, which can introduce bias when alternative scenarios exist. The method proposed in this study offers several benefits. First, it can scan the genome systematically (hypothesis-free) and identify genes whose transcripts are related. Second, because it uses SNPs (eQTLs) to test the association between two gene transcripts, it is undisturbed by the impact of confounding environmental factors that cause spurious associations. Finally, it relies on summary association statistics that are publicly available. As such, it provides a convenient path for researchers who wish to combine data from several studies to identify GRNs with higher statistical power.

By applying the devised workflow to eQTL data for blood, through the discovery and validation steps, 597 gene-pairs that aggregated into gene regulatory networks were detected. Examination of the gene functions indicated their shared involvement in immune processes, which is expected given that the eQTL data were obtained from studies using blood samples. The network topology indicated a scale-free network [19]. A core of 10 genes accounted for 20% of the interactions, and the distribution of interactions per gene followed a power-law distribution (Appendix A). If this characteristic is observed in other cell types, it would provide a convenient path for therapeutic interventions, as a scale-free network is manageable by targeting its hub genes. In this regard, IKZF1, NFE2, and CREB5 are notable because they share a higher number of SNPs with blood traits compared to other genes, suggesting that targeting them could have a broader impact (Appendix A). The findings from this study support the results of Li et al. [20], in which, based on ChIA-PET data, the authors reported that interactions between proximal and distal regulatory regions interweave into organized network communities enriched in specific biological functions.

The hub gene, *DDAH2*, exerted pleiotropic impact on several metabolic and autoimmune disorders. The metabolic function of DDAH2 is known. DDAH2 encoded protein is an enzyme that is evidently involved in production of nitric oxide by degrading asymmetrical dimethylarginine (ADMA). Previous studies indicated higher level of ADMA causes insulin resistance and T2D [21], whereas higher expression of DDAH2 improves glucose-stimulated insulin secretion [22]. Similar effects have also been reported with regard to the impact of DDAH2 and its substrate on cholesterol level. Elevated plasma ADMA levels have been observed in patients with hypercholesterolemia, atherosclerosis, and hypertriglyceridemia [23,24,25]. It has been reported that ADMA impairs cholesterol efflux in macrophage foam cells while DDAH2 overexpression prevents the detrimental effects of ADMA on cholesterol efflux [23]. The role of DDAH2 in autoimmune disorders could be indirect. Nitric oxide is vital for regulating endothelium, suppressing inflammation, and regulating immune cell function [26]. It is also reported that DDAH2 translocates to mitochondria during viral infections, where it promotes mitochondrial fission and modulates the innate immune response [27]. The antagonistic pleiotropic impact of DDAH2 expression level on disorders such as multiple sclerosis and type 2 diabetes indicate finding an optimal level is important in regulating the expression of DDAH2 for therapeutic purposes.

In this study, it was observed that a gene in the network is under a regulatory impact of about 39 eQTLs. This suggests that methods such as Mendelian randomization, which rely on SNP effects, are well suited to detect functional interactions. However, to properly examine the association between two genes, access to complete GWAS summary statistics data is required. This is especially important considering that two genes could be on different chromosomes and under the influence of distinct trans-regulatory elements. Access to complete GWAS summary statistics is also crucial for computing gene expression heritabilities. In this research, a positive correlation was noted between higher values of gene expression heritability and the likelihood of a gene exerting regulatory impact on the network. Furthermore, given that trans-eQTLs underlying the network displayed small effect sizes, constructing a GRN using eQTLs requires data from a study with decent sample size.

This study is not without limitations, although eQTLGen consortium has a large sample size and, as such, is well powered for detecting eQTLs; however, the authors did not provide complete eQTL summary association statistics, which limits the application of the current data for genome-wide studies. With regard to trans-eQTLs, the authors provided summary association statistics for 10,317 SNPs that previously showed association with the phenome. This could introduce a selection bias if SNPs are selected with regard to a specific category of traits. However, as summarized in Appendix A, this was not the case in the eQTLGen study. Namely, the selection of SNPs for trans-eQTL mapping was not with regard to a specific category of traits (Appendix A). The INTERVAL study reported their trans-eQTLs without considering their associations with phenome. Therefore, by selecting SNPs that display both cis and trans-eQTL effects and conducting Mendelian randomization, I investigated biological processes among gene-pairs from the INTERVAL study that showed significant association (*p* < 5 × 10^−8^) following Mendelian randomization. The outcome of GO-BP enrichment analysis indicated the identified genes mainly share immune function (Appendix A). Therefore, the selection procedure applied to trans-eQTL data at the eQTLGen study is unlikely to be a major issue.

In summary, this study provides a workflow based on eQTLs to identify tissue-specific gene regulatory networks. The identified network displayed scale-free topology. If further research substantiates this finding, then in each network, targeting the hub genes could provide a solution to treat abnormalities at the cellular level. Furthermore, considering that the current well-powered studies to identify trans-eQTLs are limited to blood, cataloging trans-eQTLs in other tissues is recommended.

## Figures and Tables

**Figure 1 cimb-47-00594-f001:**
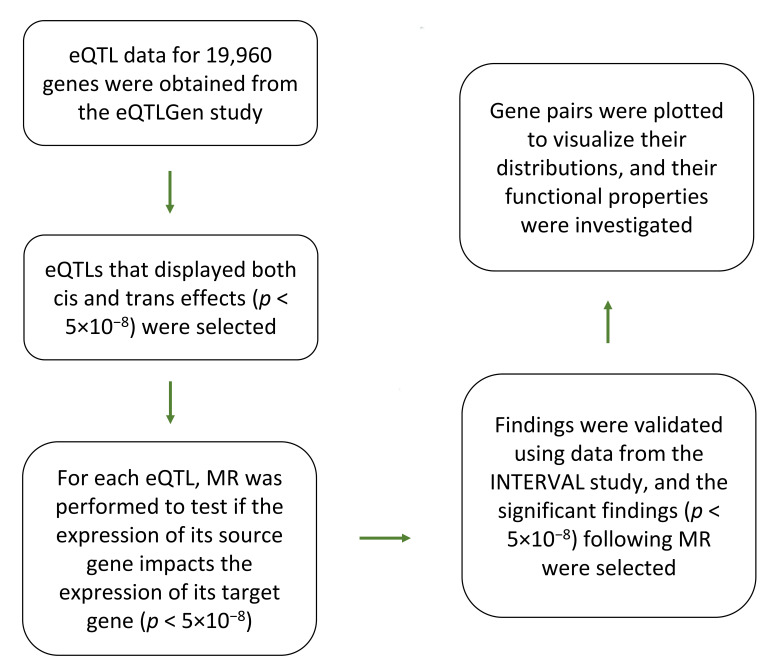
**eQTL-based analytical pipeline that was used to identify and characterize gene regulatory networks in blood.** Initially, eQTLs (*p* < 5 × 10^−8^) that display both trans and cis effects were selected from the eQTLGen study. Next, Mendelian randomization (MR) was performed to examine if change in the expression of the source gene (associated with the cis effect) impacts (*p* < 5 × 10^−8^) the expression of the target gene (associated with the trans effect). The gene-pairs obtained from this step were once more validated using the eQTL data from the INTERVAL study. Significant gene-pairs were selected and plotted to view the nature of relation between them. Functional analysis was performed on the generated network to identify significantly enriched biological processes. The properties of eQTLs underlying the network were investigated for molecular insight.

**Figure 2 cimb-47-00594-f002:**
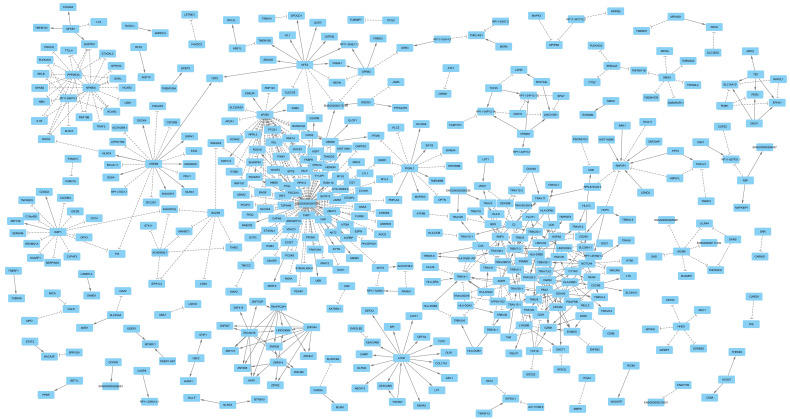
**Overview of the gene networks identified in this study.** Mendelian randomization revealed 597 gene-pairs (*p* < 5 × 10^−8^) that aggregated into networks with scale-free topology (Appendix A). An edge with an arrow end indicates that as the expression of the source gene increases, the expression of the target gene increases as well; an edge with a T end indicates an inverse association. Statistical details on the nature of association between gene-pairs are provided in Appendix A.

**Table 1 cimb-47-00594-t001:** Biological processes that are overrepresented among the identified genes.

GO-BP ID	Description	*p*	Corrected *p* ^1^
0002250	Adaptive immune response	1.0 × 10^−25^	2.3 × 10^−22^
0002504	Antigen processing and presentation of peptide or polysaccharide antigen via MHC class II	3.2 × 10^−9^	7.2 × 10^−6^
0002376	Immune system process	1.5 × 10^−8^	3.4 × 10^−5^
0002503	Peptide antigen assembly with MHC class II protein complex	1.5 × 10^−8^	3.4 × 10^−5^
0009617	Response to bacterium	1.7 × 10^−8^	3.9 × 10^−5^
0019882	Antigen processing and presentation	2.2 × 10^−8^	4.9 × 10^−5^
0007166	Cell surface receptor signaling pathway	2.8 × 10^−7^	6.4 × 10^−4^
0045087	Innate immune response	3.6 × 10^−7^	8.1 × 10^−4^
0019886	Antigen processing and presentation of exogenous peptide antigen via MHC class II	2.6 × 10^−6^	5.9 × 10^−3^
0006955	Immune response	2.8 × 10^−6^	6.2 × 10^−3^

^1^ Corrected for the Bonferroni procedure.

**Table 2 cimb-47-00594-t002:** *DDAH2* transcript level has causal impacts on several autoimmune and metabolic disorders.

Trait	PMID	Panel ^2^	B	SE	*p*	N_SNP_
Multiple sclerosis	31604244	D	1.49	0.06	2.7 × 10^−117^	9
		V	1.64	0.07	5.3 × 10^−134^	28
LDL	32493714	D	−0.08	0.01	6.7 × 10^−39^	12
		V	−0.10	0.01	1.0 × 10^−39^	20
Type 1 diabetes	34127860	D	−1.02	0.05	6.8 × 10^−88^	11
		V	−0.25	0.04	8.2 × 10^−9^	30
Rheumatoid arthritis	24390342	D	0.76	0.08	4.6 × 10^−24^	10
		V	0.60	0.07	3.7 × 10^−20^	27
Psoriasis	UKBB ^1^	D	−0.01	0.001	3.9 × 10^−22^	12
		V	−0.01	0.001	3.3 × 10^−19^	29
Primary biliary cholangitis	22961000	D	−0.81	0.10	8.2 × 10^−15^	7
		V	−0.96	0.11	1.1 × 10^−19^	24
Crohn’s disease	26192919	D	−0.25	0.04	4.8 × 10^−12^	8
		V	−0.36	0.04	1.3 × 10^−17^	23
High cholesterol	UKBB	D	−0.02	0.002	8.5 × 10^−12^	12
		V	−0.02	0.002	3.9 × 10^−17^	29
Type 2 diabetes	30297969	D	−0.11	0.02	1.6 × 10^−9^	11
		V	−0.12	0.02	1.0 × 10^−8^	29
Triglycerides	24097068	D	−0.07	0.01	3.1 × 10^−9^	8
		V	−0.07	0.01	4.9 × 10^−8^	25

^1^ UK Biobank data (https://www.nealelab.is/uk-biobank, accessed on 22 July 2025). ^2^ Source of eQTL data. D indicates discovery panel (eQTLGen study). V indicates validation panel (INTERVAL study).

## Data Availability

eQTL summary association statistics were obtained from the eQTLGen (https://www.eqtlgen.org/phase1.html, accessed on 22 July 2025) and INTERVAL study (https://www.omicspred.org/downloads, accessed on 22 July 2025). 1000 Genomes genotype data (phase 3) were obtained from https://www.cog-genomics.org/plink/2.0/resources#phase3_1kg, accessed on 22 July 2025.

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
