# Peer review of "Trans-eQTLs Can Be Used to Identify Tissue-Specific Gene Regulatory Networks"

_cimb, 2025, doi:10.3390/cimb47080594_

Round 1
Reviewer 1 Report
Comments and Suggestions for Authors
In the manuscript named “trans-eQTLs Can Be Used to Identify Tissue-Specific Gene Regulatory Networks”, author has performed trans-eQTL analysis to identify tissue-specific gene regulatory networks, and author has validated these findings using INTERVAL data. In addition, the Mendelian randomization is also performed in this manuscript. These findings would be helpful for determining human genes function in future. However, there were some comments about it.
(1) The method sections were not clearly, in section 2.1, most words were described ref11, or author method section? I am not sure whether meta-analysis was used as a method in this manuscript, nor did I know the specific parameters.
(2) In section 2.2, I didn’t know the validation work. The eQTLs was also present in INTERVAL? How many eQTL here? And was there a statistical analysis for this conclusion?
(3) In analysis process, author had used “eQTLs (P<5e-8)” in many times, why did author select this parameter? Please clearly describe.
(4) The context in this manuscript was poor, these results were not substantial enough to meet the standards of a full research paper. Would it be possible to convert this work into another article type, such as a "Comments" or "Short Communication"?
(5) Author had used “I identified …”, “I obtained …”, “I used the …”, etc, please check them through whole manuscript.
(6) Many data were collected from blood samples, how to identify tissue-specific GRN?
(7) In GO analysis, many GO terms enrichment were be interrelated Immune response, which was popular terms in blood tissue, how to confirm author analysis correctly working?
Reviewer 2 Report
Comments and Suggestions for Authors
This manuscript makes a valuable contribution to bioinformatics, particularly in the field of gene regulatory network (GRN) analysis. The author proposes a practical, modern approach to reconstructing regulatory networks that does not require individual-level data, instead using publicly available eQTL data and Mendelian randomisation. This is a notable strength of the work, particularly given concerns about data privacy and the reusability of large-scale resources.
The methodology is clearly described and well structured. Using two independent datasets (eQTLGen and INTERVAL) strengthens the robustness of the findings. Identifying scale-free network topology, where a subset of hub genes exerts widespread regulatory influence, aligns with known biological principles and offers opportunities for targeted therapeutic strategies. Functional enrichment analyses confirm that the identified genes are primarily involved in immune-related processes, consistent with the use of whole-blood data.
The author openly discusses the study’s limitations, which include:
- lack of access to the full summary statistics from the eQTLGen Consortium, which limits broader genome-wide analysis;
- modest effect sizes of trans-eQTLs, necessitating large sample sizes for reliable detection;
The analysis was also limited to one tissue (blood), which restricts generalisability across tissues.
My comments:
- It is recommended that the discussion be expanded to encompass concise commentary on the potential clinical implications of the identified hub genes. For instance, genes such as IKZF1, NFKBIA, and NFE2 are well-known for their roles in haematopoiesis and immune regulation, and thus may represent promising therapeutic targets.
- The process of biological validation is a critical component of scientific research, involving the use of biological materials to verify the accuracy and reliability of experimental results.
- It is recommended that the manuscript be enhanced by the incorporation of literature-based validation for a minimum of one gene pair that has been identified through the analysis process. This will serve to provide a more robust substantiation of the biological relevance of the findings.
- It is imperative to maintain uniformity in the placement of references and the formatting of citations. In the current version of the text, references appear mid-sentence without clear contextual linkage.
Comments on the Quality of English Language
- Occasionally, the manuscript exhibits characteristics of colloquial language, as illustrated by the use of informal contractions such as "I used" and "I obtained". It is recommended that an impersonal, scientific style be employed consistently throughout the text (e.g., "Data were obtained", "The analysis was conducted…").
- It is recommended that a language edit be undertaken in order to enhance fluency, eradicate repetitive phrasing (for example, "expression of the source gene impacts expression of the target gene") and rectify minor grammatical issues (for example, "attribute" → "attributed").
Round 2
Reviewer 1 Report
Comments and Suggestions for Authors
Thanks for authors’ works, the manuscript had been well revised, most of my comments were well addressed in revision. But the context of manuscript was still poor, and although author had added many words, but the work was poor as research article. In addition, the manuscript was still needed to improve writing before publishing. Good luck.
Author Response
Comments 1: The context of manuscript was still poor, and although author had added many words, but the work was poor as research article. In addition, the manuscript was still needed to improve writing before publishing.
Response 1: The amount of data and the computational time used to generate the results are considerable. Furthermore, the study used a validation step and simulation to minimize the likelihood of bias. These are the strengths of the study.
The manuscript has been also revised with regard to the language.